# Current Adenosinergic Therapies: What Do Cancer Cells Stand to Gain and Lose?

**DOI:** 10.3390/ijms222212569

**Published:** 2021-11-22

**Authors:** Jana Kotulová, Marián Hajdúch, Petr Džubák

**Affiliations:** Institute of Molecular and Translational Medicine, Faculty of Medicine and Dentistry, Palacký University Olomouc, 779 00 Olomouc, Czech Republic; jana.kotulova@upol.cz (J.K.); marian.hajduch@upol.cz (M.H.)

**Keywords:** adenosine, adenosine receptors, cancer, adenosinergic therapy, tumour microenvironment, immunosurveillance, adverse effects, immuno-oncology

## Abstract

A key objective in immuno-oncology is to reactivate the dormant immune system and increase tumour immunogenicity. Adenosine is an omnipresent purine that is formed in response to stress stimuli in order to restore physiological balance, mainly via anti-inflammatory, tissue-protective, and anti-nociceptive mechanisms. Adenosine overproduction occurs in all stages of tumorigenesis, from the initial inflammation/local tissue damage to the precancerous niche and the developed tumour, making the adenosinergic pathway an attractive but challenging therapeutic target. Many current efforts in immuno-oncology are focused on restoring immunosurveillance, largely by blocking adenosine-producing enzymes in the tumour microenvironment (TME) and adenosine receptors on immune cells either alone or combined with chemotherapy and/or immunotherapy. However, the effects of adenosinergic immunotherapy are not restricted to immune cells; other cells in the TME including cancer and stromal cells are also affected. Here we summarise recent advancements in the understanding of the tumour adenosinergic system and highlight the impact of current and prospective immunomodulatory therapies on other cell types within the TME, focusing on adenosine receptors in tumour cells. In addition, we evaluate the structure- and context-related limitations of targeting this pathway and highlight avenues that could possibly be exploited in future adenosinergic therapies.

## 1. Introduction

Adenosine (ADO) is an omnipresent and rapidly metabolized purine nucleoside with a physiological half-life of a few seconds [1]. Concentrations of circulating ADO in vivo are therefore challenging to measure [2]. Physiological concentrations of extracellular ADO (eADO) have been reported to be in the low nanomolar range, but under pathological conditions, they can be as high as 100 mM [3,4,5,6]. Given its instability, ADO primarily acts via autocrine and paracrine signalling. It is involved in cellular energy transfer because it is a building block for the formation of adenosine diphosphate (ADP) and adenosine triphosphate (ATP). Additionally, it plays important roles in various signal transduction pathways as a component for the formation of signalling molecules such as cyclic adenosine monophosphate (cAMP).

The primary source of eADO is molecules of ATP that are released uncontrollably as a result of physical damage, exposure to various stress stimuli, or deliberate non-lytic ATP efflux. ATP is hydrolysed to ADO by a series of membrane-localised enzymes in several cell types, in particular ecto-nucleoside triphosphate diphosphohydrolase-1/CD39, ecto-5′-nucleotidase/CD73, ectonucleotide pyrophosphatase/phosphodiesterase (ENPP), and prostatic acid phosphatase (PAP). Generation of eADO is also possible via intrinsic metabolic pathways mainly involving adenosine kinase (ADK), S-adenosylhomocysteine hydrolase (SAHH), cytoplasmic 5′-nucleotidase-I (cN-I), and the NAD^+^ salvage pathway via cyclic ADP ribose hydrolase (CD38) on the cellular surface [7,8]. Interestingly, SAHH and the nuclear isoform of ADK (ADK-L) regulate the transmethylation pathway by controlling nuclear ADO levels. ADK-L limits the availability of ADO in the nucleus and thus augments DNA and histone methylation and subsequent epigenetic changes [9]. The cellular uptake of ADO is mediated by bi-directional equilibrative nucleoside transporters (ENTs) and one-way concentrative nucleoside transporters (CNTs) that also modulate the efficacy of administered drugs [10]. Extracellular and intracellular ADO availability are further limited by ADO catabolism. In particular, adenosine deaminase (ADA) metabolizes ADO to inosine, and purine nucleoside phosphorylase (PNP) catalyses the conversion of inosine to hypoxanthine or other purines [11], with both reactions occurring on both sides of the cellular membrane (Figure 1).

ADO was originally described as a cardiovascular system modulator in 1929 [12] and its primary extracellular targets, adenosine receptors (ARs), were first described in the 1970s. It has since become apparent that ADO and four ARs (A_1_R, A_2A_R, A_2B_R and A_3_R) play roles in a number of pathological conditions, including cancer. The effects of the adenosinergic pathway in immune cells were recently reviewed by other research groups [13,14,15]. Here, we focus on what is currently known about the adenosinergic system with emphasis on its benefits and disadvantages for cancer cells.

## 2. Fine-Tuned Orchestration of the Adenosinergic Pathway in Cancer

In addition to their role in intercellular communication, cues from the extracellular matrix (ECM) reciprocally influence the healthy tissue architecture and the associated tissue-specific functions; this process is referred to as bidirectional tissue microenvironment dynamic reciprocity [16]. After the initial transformation occurs in the precancerous niche, molecular heterogeneity increases as mutations accumulate, creating clonal diversity in the local cellular population. This creates heterogeneity in metabolic interactions, allowing unfavourable conditions within the tumour microenvironment (TME) to be overcome. Cells of all types within the TME face several challenges including elevated interstitial pressure, growing demands for oxygen and nutrients, impaired supplies delivery, and inefficient metabolite clearance. Reciprocally, changes in cellular metabolism within the TME may direct interactions with the ECM in ways that support the tumour’s sustenance and accelerate the remodelling of the TME. This process is mediated via shared cues such as nutrients, metabolites, amino acids, fatty acids, macromolecules, small peptides, organelles, and nucleotides and nucleosides. The combination of these intrinsic and extrinsic factors and associated inflammation, hypoxia, and oxidative stress ultimately lead to solid tumour progression (summarised in [17]).

It has been shown that the release of purines into the extracellular environment plays an important role in intercellular communication [18]. Indeed, the pericellular release of ATP as a ‘danger signal’ is a key part of the damage-associated molecular pattern (DAMP) signalling system [19], a highly evolutionary conserved mechanism for coping with tissue damage. Accordingly, most cell types express the full assortment of ADO producing and metabolizing enzymes at various levels [20,21]. Under physiological conditions, ADO’s tissue-protective and anti-nociceptive effects counterbalance the pro-immunogenic and pro-inflammatory activity of ATP [22].

In the TME, the immunogenic properties of fellow cancer cells dying following tissue damage or chemotherapy as well as deliberate ATP release are neutralized by ATP hydrolysis to ADO [23,24]. Interestingly, chemotherapeutic drugs differ in the extent to which they induce ADO accumulation in the extracellular space [25]. However, it is not only cancer cells that release eADO to maintain specific immunoinhibitory phenotype [26]; indeed, dying T regulatory (Treg) cells within the TME provide both ATP molecules and CD39/CD73 ectoenzymes to sustain the ADO-rich TME, which in turn triggers immunosuppression in adjacent effector cells via A_2A_R [27]. Another interesting positive feedback loop was observed in infiltrating neutrophils, where ENT expression is depressed via a hypoxia-inducible factor 1α (HIF-1α)-dependent mechanism, ensuring that high levels of ADO are maintained in the extracellular space [28]. Furthermore, the pharmacological and genetic blocking of ENT1 led to an increase in eADO levels and to subsequent activation of A_2A_R and A_2B_R in the acute lung injury murine model [29]. If ENT1 remained functional, the detrimental inflammatory response was triggered. Interestingly, the deactivation of ADA in neutrophils has similar effects [30]. Moreover, large numbers of cancer-associated fibroblasts (CAFs) are present in the TME [31]. In colorectal cancer, CAFs were shown to maintain elevated ADO concentrations in the TME via high expression of CD73 driven by A_2B_R stimulation [32]. These results show that the adenosinergic pathway is hijacked by the TME population and that all cell types within the TME (cancer, stromal, endothelial, and immune cells) are affected by ADO via AR-dependent and independent routes. Thus, ADO, which under physiological conditions serves to alleviate immune system overreaction and prevent tissue damage, ultimately becomes an agent that supports unrestricted tumour growth.

## 3. Current Therapeutic Focus

Recent landmark breakthroughs in research on anti-tumour immune response [33,34] have led to a shift of attention away from ARs on cancer cells to the metabolism of ADO and its role as an endogenous agonist of ARs on immune cells in the tumour niche [7,8,17,21,35,36]. These findings prompted novel pre-clinical and clinical therapeutic approaches targeting different adenosinergic signalling components and the whole purinome. Notable targets in these efforts have included CD39 [37,38], CD73 [13,39], CD38 [40], and A_2A_R [41,42,43]. Other studies have focused on chimeric antigen receptor T (CAR-T) cells with A2AR suppression to block T-cell functionality [44,45] and on combination therapies targeting the above-mentioned proteins together with anti-programmed cell death protein 1 (PD-1), anti-programmed cell death protein ligand 1 (PD-L1) [46] or anti-cytotoxic T-lymphocyte-associated protein 4 (CTLA-4) [47]. Whether used as monotherapies or in combination with other therapeutic agents, current immune-oncological treatments aim to increase the immune response of the host immune system against the tumour mass, mainly by inhibiting ATP hydrolysis to ADO or AR signalling in immune cells. This is illustrated by the therapeutic targets of recently initiated clinical trials involving the adenosinergic pathway (Table 1). 

Another important discovery is that the increased levels of ADO are associated with poor prognosis in patients, suggesting that ADO could serve as a tractable prognostic biomarker. Accordingly, a gene expression signature analysis revealed a positive role of ADO in promoting cancer by boosting transforming growth factor β (TGF-β) signalling and antagonizing anti-PD-1 therapy [48]. More recently, an ADO gene signature (AdenoSig) was identified in patients with renal cell cancer, consisting of immune-related genes encoding interleukin 1β (IL-1β), prostaglandin-endoperoxide synthase 2 (PTGS2), and C-X-C motif chemokine ligand 1 (CXCL1), 2, 3, 5, 6, 8 [46].

A potential issue with all immunotherapeutic strategies is their capacity to influence both tumour cells and adjacent cells within the altered niche as well as cells of the immune system. In the following text, we summarise recent discoveries concerning the effects of ARs and therapies targeting them on different cell types in tumours, with particular emphasis on cancer cells. 

## 4. Targeting ARs on Cancer Cells

ARs belong to a large family of G protein-coupled receptors (GPCRs) known as P1 receptors [49]. The A_1_R receptor has similarities with A_3_R, while A_2A_R is more closely related to A_2B_R [50]. All ARs are endogenously activated by eADO and mediate its protective function in response to stress stimuli, tissue damage, or inflammation. ARs are canonically considered to be coupled to Gi (A_1_R, A_3_R) and Gs (A_2A_R, A_2B_R) protein subunits that respectively inhibit and activate adenylyl cyclase (AC). AC catalyses the conversion of ATP to cAMP, which in turn participates in a chain of downstream signalling processes [51]. However, mechanistic studies on signal transduction in the A_1_R-A_2A_R heterotetramer have provided new insights into the coupling of ARs to Gi and Gs proteins and suggest that some caution may be needed when interpreting previous results [52]. ARs can also be classified based on their affinity for their endogenous agonist ADO: A_2A_R and A_1_R are high-affinity ARs, while A_2B_R and A_3_R require higher concentrations of eADO for activation and are thus called low-affinity ARs [18,53]. 

A_1_R is the most highly conserved member of the AR family. Townsend-Nicholson et al. mapped the *ADORA1* gene in 1995 to the human chromosomal locus 1q32.1 [54]. A_1_R is widely expressed, mainly in the central nervous system as well as in the peripheral nerves and heart [50]. Accordingly, it is involved in neurotransmission and neuromodulation. Additionally, it has been reported that A_1_R receptor stimulation protects the brain and heart tissue against ischemic/reperfusion damage [53]. The human A_2A_R receptor gene *ADORA2A* is localized on chromosomal locus 22q11.23 [55]. A_2A_R is abundantly expressed in the leukocytes, platelets, spleen, thymus, and striatopallidal neurons; at lower levels in the heart, blood vessels, and lungs [50]. In addition, A_2A_R is involved in regulating heart rhythm and blood flow, ischaemic preconditioning of the heart and brain [53], and immune reaction attenuation [56]. The function of A_2B_R has been elusive for a long time because of a lack of specific ligands. However, the gene encoding human *ADORA2B* is located on chromosome 17p12-p11.2 [54]. A_2B_R is widely expressed throughout the human body, albeit mostly at low levels. Its expression is strongest in the gastrointestinal tract, bladder, on the surface of the mast cells and in the lungs; it is also expressed to a lesser degree in the brain, kidney and adipose tissue [50]. The A_2B_R receptor was proposed to protect tissue against the detrimental effects of inflammation, hypoxia or ischemia. Unlike other ARs, it is activated by micromolar levels of ADO in the environment [57]. The most recently discovered AR gene, *ADORA3*, encodes A_3_R and is localized on chromosome 1 at 1p13.2 [58]. It was cloned and pharmacologically characterized in 1993 by Salvatore et al. High levels of A_3_R are predominantly observed in lung and liver tissue [59].

Previous studies have shown that ARs are attractive but challenging therapeutic targets [60]. Reflecting the complexity of adenosinergic signalling, current approaches targeting ADO metabolism and ARs have impacts at multiple distinct levels within the TME, as shown in Figure 2. Herein, we summarise recent findings concerning the direct and indirect effects of targeting ARs in the tumour niche on various aspects of tumorigenesis.

### 4.1. Cancer Cell Proliferation

ARs are expressed at high densities in tumours [13,61,62,63], and high A_1_R expression is correlated with lower overall survival in hepatocellular carcinoma (HCC) patients [64]. Moreover, in vitro and animal experiments showed that A_1_R overexpression promoted cancer cell proliferation via the phosphoinositide-3-kinase (PI3K)/AKT pathway and that treatment with the specific A_1_R antagonist DPCPX suppressed tumour progression in hepatocellular carcinoma. A_1_R overexpression was also associated with tumorigenesis and an invasive profile via the PI3K/AKT/glycogen synthase kinase 3β (GSK-3β)/β-catenin pathway in nasopharyngeal carcinoma [65]. Treatment with a dual inhibitor of A_1_R and ornithine decarboxylase 1, ODC-MPI-2, increased cAMP levels and reduced polyamine production, ultimately leading to growth inhibition in triple-negative breast cancer (TNBC) cell lines [66]. Although the molecular mechanism underpinning this response remains to be elucidated, it shows the potential of combination therapies targeting the AR-cAMP pathway and polyamine synthesis.

Activated A_2A_R stimulates the cAMP/PI3K/AKT proliferative pathway directly in tumour cells [67,68] but a number of promising agents have shown indirect anti-tumour effects mediated via A_2A_R on immune cells [69]. For example, A_2A_R antagonists such as AZD4635 [42], AB928 [70], and ciforadenant [71] are currently being studied to evaluate their beneficial effects on immune responses in tumours. The results available at present suggest that tumour growth is reduced by restoring immunosurveillance via A_2A_R antagonism and that, furthermore, tumour cell proliferation is directly inhibited by simultaneous blocking of A_2A_R on tumour cells. 

A_2B_R was identified as a critical factor for proliferation in head and neck squamous cell carcinoma (HNSCC) cell lines; its inhibition led to a reduction in intracellular cAMP production, cell cycle arrest in the G1 phase, and induction of apoptosis in vitro as well as a reduction in tumour growth and vascularisation in vivo [72]. The A_2B_R subtype also seems to be upregulated in renal cell carcinoma (RCC) and their blockage via pharmacological intervention using MRS1754 or via shRNA knockdown suppressed RCC cell proliferation and migration [73]. In addition, A_2B_R downregulation using shRNA blocked tumour growth in vivo. Interestingly, treatment with an A_2B_R agonist rescued cells from the effects of antagonist treatment via a pathway involving the stress-induced mitogen-activated protein kinase (MAPK) JNK.

Although blocking A_2B_R has predominantly anti-proliferative effects in several tumour types, other reports indicate that activated A_2B_R can have anti-tumour effects in cancer. In the MBA-MD-231 TNBC cell line, treatment with A_2B_R agonists attenuated three distinct signalling pathways: cAMP, Ca^2+^ and extracellular signal-regulated kinase 1/2 (ERK1/2), with ERK1/2 signalling being most strongly downregulated [74,75]. Therefore, blocking of A_2B_R in immune cells could potentially trigger their proliferation. However, it can be challenging to accurately assess the degree of bias signalling when using different experimental methods designed to analyse different levels within the hierarchy of a signalling cascade [76]. Moreover, any observed bias could potentially be due to a bias of the receptor itself or to specific responses of the systems surrounding the GPCR rather than any bias of the agonist per se [77]. An interesting mechanism linking metabolic changes to tumour cell death was described by Long et al. [25]: ARs are activated by elevated eADO levels but ADO accumulation also activates the well-known wild-type tumour protein p53 (TP53), which in turn upregulates A_2B_R expression. In the presence of a suitable agonist, the resulting A_2B_R activation leads to cancer cell death in a PUMA-dependent manner. In addition to its interactions with TP53, eADO can also interact with a paralog of TP53, TP73, in tumours where TP53 is mutated or debilitated. Stimulation of the TP73-A_2B_R axis induces caspase-dependent apoptosis in cells exposed to chemotherapeutics [78]. This could be especially useful since many tumour types harbour mutations in TP53. Moreover, TP53 mutations in melanoma patients were associated with increased CD73 expression, which in turn correlated with the metastatic potential of the melanoma [79]. This indicates that adenosinergic pathways are entwined with TP53 on multiple levels.

Although A_3_R exhibits high homology with A_1_R and thus has similarities in its downstream pathways, its agonists mainly mediate anti-proliferative effects. For example, the specific A_3_R agonist 2-Cl-IB-MECA induced cell cycle arrest and inhibited melanoma cell proliferation and lung metastasis in a murine model [80]. Later studies using other tumour models revealed that A_3_R activation resulted in downregulation of the cAMP-dependent PI3K/AKT axis and ERK1/2 kinase [81], nuclear transcription factor-κB (NF-κB), and the Wnt/β-catenin pathway [82]. These findings led to the testing of A_3_R agonists in clinical trials against HCC [83].

### 4.2. Hypoxia and Immunomodulation

Insufficient oxygenation is a common feature of solid tumours and hypoxia is a supportive mechanism for the immunodeficient TME [84]. The immunosuppressive [33], pro-tumour [85] and pro-metastatic [86] role of HIF-1α is well established. Its link to ARs, however, is relatively new [87,88]. The strongest evidence linking hypoxia to ARs derives from studies on A_2A_R in immune cells. In an ADO-rich hypoxic milieu, activation of A_2A_R expressed on T effector (Teff) cells upregulated the cAMP-protein kinase A (PKA)-cAMP-response element-binding protein (CREB) pathway that assists HIF-1α in promoting the transcription of target genes including TGF-β, IL-10, and CD39/CD73. A_2A_R and HIF-1α are also present and activated in Treg cells, where they further attenuate Teff cell responses [89]. A similar mechanism involving the adenosinergic pathway was shown to incapacitate other immune cells [13,33]. Moreover, the cAMP-PKA-HIF-1a-CD39/CD73-ADO-A_2A_R loop in HeLa cervical cancer cells and rat cardiomyocytes was shown to contribute to the pathological changes [90]. 

A notable hypoxia-related morphogen is vascular endothelial growth factor (VEGF). In accordance with the immunomodulatory role of A_2A_R in TME, the A_2A_R agonist polydeoxyribonucleotide exhibited anti-inflammatory effects in an ischemic colitis rat model, suppressing the expression of pro-inflammatory cytokines and increasing levels of A_2A_R and VEGF. This in turn suppressed the adverse effects of mucosal damage and promoted healing of ischemic tissue [91]. Analogously, ADO stimulates HIF-1α and VEGF production and VEGF secretion by human macrophages via activation of A_2A_R [92]. Elevated levels of the pro-inflammatory and protumoral VEGF have been detected in dendritic cells (DCs), tumour-associated macrophages, and Treg cells as well as in cancer cells, and high expression of A_2A_R in RCC correlates with metastases in patients [63]. All these factors suggest that A_2A_R may be an attractive anticancer therapy target. In primary tumours, the comparatively low protein-level expression of A_2A_R resulted in a better response of the patients to anti-VEGF and immune checkpoint inhibitor (ICI) therapy. However, anti-VEGF treatment had no apparent effect on PD-L1 expression. In addition, simultaneously elevated CD73 and A_2A_R expression led to shorter overall survival, indicating that eADO production is important for the A_2A_R activity that enables evasion of the host immune response. It should be noted, however, that therapeutically-induced systemic oxygenation alone was sufficient to diminish the hypoxia-induced HIF-1α- and CD39/CD73-driven ADO-enrichment of the TME and the expression of A_2A_R and A_2B_R, leading to the restoration of immunosurveillance [87]. 

Being under transcriptional control of HIF-1α, A_2B_R also plays an important role in hypoxia [93]. In a sepsis mouse model, the anaesthetic sevoflurane recently had immunomodulatory effects that depended on A_2B_R expression, further supporting the notion that A_2B_R is tightly linked to HIF-1α [94]. Further, A_2B_R was shown to stimulate the production of a cocktail of pro-angiogenic, pro-inflammatory, and immunosuppressive mediators including VEGF, IL-8, IL-6, IL-10, cyclooxygenase-2, TGF-β, and indoleamine 2,3-dioxygenase (IDO) in differentiated DCs under hypoxic conditions in the ADO-rich TME [95]. There is also evidence of similar signalling pathway activation in cancer cells. For example, in breast cancer stem cells (CSCs), hypoxia stimulates A_2B_R expression via activation of HIF-1a transcriptional activity. A_2B_R in turn helps maintain the dedifferentiated phenotype of breast CSCs by promoting IL-6 and NANOG expression. From a therapeutic perspective, it is notable that curtailing the expression and activity of A_2B_R reduced tumour growth and metastatic dissemination in vivo [96]. In addition, a CD73-A_2B_R-dependent increase in IL-10 production reduced the surface expression of major histocompatibility complex class I molecules (HLA-I) on cervical cancer cells, rendering CD8+ Teff cells unable to recognize and engage them [97]. 

On the other hand, the selective A_2B_R antagonist PSB-603 reduced inflammatory responses by downregulating the pro-inflammatory cytokines IL-6 and tumour necrosis factor α (TNF-α), and also reduced reactive oxygen species (ROS) levels in a murine model of local and systemic inflammation, where it also blocked the recruitment of leukocytes to the inflammation site [98]. Although the molecular mechanism driving this response was not investigated, the report suggests a potentially ambiguous role of A_2B_R in TME. Another study showed that hypoxia drove TME-associated cells toward metabolic reprogramming and increased the production of immunosuppressive IDO, which in turn constrained the stimulation of Teff cells. Blocking of A_3_R inhibited IDO production while blocking of A_2B_R resulted in enhancement of IDO production in DCs to maintain the immunosuppressive phenotype [99]. IDO upregulation is linked to the PD-1/PD-L1 pathway because it has a downstream role in PD-1/PD-L1 signalling [100]. These reports suggest that A_2B_R inhibition therapy could support the development of tumour-tolerant DCs, especially under hypoxic conditions. However, the currently available evidence indicates that A_1_R and A_3_R play only marginal roles in hypoxia.

### 4.3. Migration and Angiogenesis

A growing body of evidence suggests that there is a link between angiogenesis, motility and the adenosinergic pathway [101]. In endothelial progenitor cells (EPC), expression of the C-X-C motif chemokine receptor 4 (CXCR4) is upregulated after ADO treatment, and ADO can subsequently increase EPC migration to the heart after myocardial infarction where it stimulates angiogenesis via A_2B_R and CXCR4-dependent mechanisms [102]. Possible effects of A_2A_R in this process were not excluded. The formation of capillary-like structures in HMEC-1 microvascular endothelial cells was promoted by A_2B_R activation, a mechanism that would occur in damaged tissue [103]. Production of VEGF required activation of the cAMP-PKA-CREB axis, while endothelial nitric oxide synthase (eNOS) induction was mediated by the PI3K/AKT pathway and both VEGF and eNOS were necessary for A_2B_R-stimulated angiogenesis. Oxidative stress is thus clearly a key modulator of endothelial cell function; whereas controlled ROS production induces angiogenesis, excessive ROS levels are detrimental for endothelial cells [104]. 

Stromal cells are important factors for tumour cell expansion and the associated angiogenesis, which is modulated by the adenosinergic pathway. TGF-β has been shown to regulate the ADO-generating enzymes CD39, and CD73, which accelerated tumour progression by promoting the maturation of myeloid-derived suppressor cells (MDSC) [105]. Vasiukov et al. [106] linked TGF-β deletion in myeloid cells to deregulation of CD73′s catalytic activity. Conversely, upon stimulation by TGF-β, myeloid cells expressing CD73 generated high levels of eADO that in turn led to TGF-β downregulation in CAFs via A_2A_R/A_2B_R-cAMP production. This gave rise to a less regimented stromal milieu, allowing ECM remodelling and the spread of cancer cells to distant locations. Accordingly, previous studies have shown that the cAMP/PKA axis regulates cytoskeletal organization and migration [107]. 

ADO-activated A_2B_R increased NADPH oxidase 2-dependent ROS production and inhibited neovascularization. In contrast, the specific A_2B_R inhibitor MRS1706 reduced oxidative stress [108]. Similarly, genetic and pharmacological inhibition of A_2A_R ultimately resulted in downregulation of ROS production and oxidative stress in normal endothelial cells, preserving their functions [109]. In contrast, treatment with an A_2A_R agonist suppressed the migration of lung adenocarcinoma cells to the brain. This is because stimulation of A_2A_R causes deregulation of the stromal cell-derived factor-1 (SDF-1)/CXCR4 axis promoting migration, and also enhances the integrity of the blood–brain barrier, thus inhibiting brain metastases in mice [110]. 

During tissue damage, exposure to eADO changes the expression pattern of adhesion molecules, thus deregulating immune cells’ attachment to the endothelium and their extravasation to the inflammation site [111]. It was recently shown that prolonged exposure to the non-selective AR agonist NECA may stimulate cancer cell movement through the endothelium and could thus contribute to metastasis; such effects were not seen following short exposure [112]. Although the ARs responsible for this outcome were not identified, these findings illustrate the spatial and temporal significance of AR targeting. It is notable that the ability of CD73 to degrade AMP into ADO is vital for the epithelial integrity of the normal endometrium. Moreover, CD73 is deficient in poorly differentiated and advanced endometrial carcinomas but acts to preserve the epithelial architecture during the early stage of tumour development [113]. Another important molecule required for cell-cell adhesion and polarity, Rap1B, is activated by the A_2A_R and A_2B_R signalling pathways in metastatic tumour models [68,114]. A mechanistic study using a reporter system revealed that targeting A_2A_R and A_2B_R could affect this GTP-binding protein via posttranslational modifications [115]. As a consequence, A_2A_R and A_2B_R antagonist treatment could help inhibit cancer cell migration.

### 4.4. Tumour Cell Stemness and Reprogramming

High-grade malignancy and metastasis are defined by the ability of tumour cells to resist apoptosis, invade, and disseminate. The epithelial–mesenchymal transition (EMT), stemness, and cellular plasticity of CSCs all contribute to poor patient outcomes [84]. EMT promoting factors such as TGF-β, Wnt/β-catenin, TWIST, and the oncogenes KRAS and epidermal growth factor receptor (EGFR) shape the TME by upregulating CD73 expression [13]. In turn, CD73 putatively promotes CSC stemness by generating eADO that activates ARs. This hypothesis is supported by the observation that treatment with the non-selective AR antagonist caffeine reduced the sphere-forming efficiency of ovarian CSCs. Furthermore, transcription of EMT drivers (Snail, TWIST1, ZEB1) and stemness genes (NANOG, OCT4, SOX2, SOX9) appears to be controlled by CD73 [116,117]. 

Multiple studies support the notion that the CD73-induced amplification of EMT cues is mediated by ARs. For example, the catalytic activity of CD73 was linked to the activation of Snail—a key molecule in the EMT [118]—and the non-selective phosphodiesterase inhibitor pentoxifylline downregulated both CD73 and the transcription factor ZEB1, another EMT activator. Noteworthy, pentoxifylline is also a xanthine derivative with non-specific antagonistic activity towards ARs [119]. Additionally, an A_3_R antagonist reduced the clonogenic potential of glioblastoma stem-like cells (GSCs) and promoted their apoptosis. GSCs are a resistant subpopulation of glioblastoma tumours characterized by increased eADO levels, especially under hypoxic conditions [120]. Moreover, A_3_R inhibition hampered EMT-associated processes in glioblastoma non-CSC cells [121], and A_3_R facilitated the EMT of GSCs, especially under hypoxic conditions that activated HIF-2 and ADO production via PAP [122]. HIF-2α and PAP are also upstream initiators of A_2B_R activation in GSC proliferation [123]. Another study identified A_2B_R as a modulator of EMT based on the balance between cAMP and MAPK pathways; according to this model, A_2B_R-MAPK activation reinforced the EMT process [124]. 

Another research group found that activation of A_1_R and A_2B_R inhibited proliferation in glioblastoma CSCs via the ERK1/2 pathway and induced apoptosis both alone and in synergy with the chemotherapeutic agent temozolomide. Interestingly, the A_1_R agonist N^6^-cyclo-hexyladenosine also induced CSCs differentiation via a mechanism involving HIF-2α [125]. In addition to reducing mammosphere formation in breast CSCs, micromolar ADO concentrations inhibit proliferation, induce apoptosis, and downregulate the activity of ERK1/2 and GLI-1 expression [126]. Later studies from the same group elaborated the mechanism responsible for these outcomes, showing that the effects of eADO on breast CSCs were mediated by activation of A_2B_R and A_3_R [127,128]. ADO is one of the external factors that promotes NANOG expression via A_2B_R-dependent activation of the PKA-IL6-STAT3 pathway, leading to activation of a rare subset of latent endogenous plastic somatic (ePS) cells [129]. Although ePS cells have a relatively low propensity for tumour formation, they possess the self-renewal capabilities and plasticity to give rise to other cell types upon NANOG activation [130]. Importantly, a specific mutation of another EMT driver, TGF-β, is associated with augmented ADO signalling and poor prognosis for patients [48]. The fact that only one TGF-β mutation has been linked to ADO-mediated tumour progression to date may imply that other cues within the tumour determine the final impact of ADO on tumorigenesis.

The ADO pathway has also been suggested to influence tumours’ resistance to various therapeutics. In glioblastoma A172 cells, γ-radiation promotes DNA damage response accompanied by an increase in cellular motility and actin remodelling. Highlighting the involvement of the CD73–ADO–A_2B_R axis, the treatment with an A_2B_R antagonist or A_2B_R siRNA knockdown downregulated the γ-radiation-related enhanced motility and actin remodelling, ultimately leading to cell death [131]. In concordance with these results, A_2B_R activation with the selective agonist BAY60-6583 increased the survival of irradiated mouse melanoma B16 cells, making A_2B_R a contributing factor to tumour radioresistance [132]. Importantly, A_2B_R blockage also suppressed EGFR translocation and its phosphorylation upon γ-irradiation, thwarting EGFR-mediated recovery of lung cancer cells from γ-radiation-stimulated DNA damage [133]. Another tumour response to radiotherapy involves HIF-1α activation to enhance the radioresistance of endothelial cells needed to sustain tumour vasculature [134]. Moreover, A_2A_R signalling enhanced the growth and invasiveness of radioresistant TNBC cells in vitro and in vivo and upregulated the expression of the EMT-related proteins Snail and vimentin [135]. Finally, there is a rare population of cancer cells called cycling persister cells that can evade therapy and proliferate under constitutive drug treatment, thus possibly contributing to tumour recurrence. Cycling persisters use non-genetic mechanisms to reprogram their metabolism toward fatty acid oxidation. Interestingly, ADO and inosine were among the metabolites upregulated after prolonged drug treatment in the cycling persistent population of metastatic PC9 cells [136].

### 4.5. Extracellular Vesicles

It is becoming increasingly apparent that extracellular vesicles (EVs) and exosomes (i.e., endosome-derived EVs) play important roles in intercellular communication as carriers of various proteins, lipids and nucleic acids, especially under stress conditions [137]. Perhaps unsurprisingly, they also carry the ADO-producing ectoenzymes CD39 and CD73 [138] and can cover large distances through the lymphatic system and blood vessels [139].

Exosomes released from tumour cells express CD39 and CD73, which increase ADO levels in the TME and suppress T cells’ functions [140]. Additionally, EVs derived from tumour cells were shown to contain ADO and inosine [141,142]. Interestingly, contact between EVs and CD8+ Teff cells caused the Teff cells to secrete perforin, leading to disruption of the EV membrane. This would release the enclosed ADO, which could then have an immunosuppressive effect on cytotoxic activity of Teff cells that could presumably hinder CAR-T therapy among other things [141]. Moreover, tumour-derived exosomes loaded with ADO, inosine, and CD39/CD73 could travel to distant tumour sites where they could trigger endothelial cell growth and polarization of macrophages towards an M2-like phenotype via activation of A_2B_R. In this way, they would simultaneously directly and indirectly promote angiogenesis [142]. However, a study by Angioni et al. describes the inhibition of tumour-associated angiogenesis, putatively via eADO [108]. Inflammation-stimulated mesenchymal stromal cells released CD39/CD73-enriched EVs producing eADO, which subsequently stimulated A_2B_R in nearby endothelial cells. A_2B_R stimulation resulted in the increased NADPH oxidase 2-dependent overproduction of ROS, which, in contrast to moderate ROS levels, is detrimental to neovascularization. 

Surprisingly, immune cells can also generate EVs carrying enzymatically active CD39 and CD73 that further dampen the immune response. A recent rodent study [143] showed that similarly to Treg cells, B cells can produce CD19+ EVs carrying CD39 and CD73, which hydrolyse extracellular ATP released from chemotherapy-treated cancer cells to suppress cytotoxic Teff cell function [144]. The generation of eADO via EVs thus appears to be a conserved immunosuppressive mechanism among immune cells. The small G proteins Rab27a and Rab27b were found to control exosome secretion in HeLa cancer cells, and a study by Ostrowski et al. also suggested Rab27a to be important for exosome formation in various cellular models [145]. Accordingly, Rab27a is critical for the genesis of CD19+ B cell-derived EVs from tumour-bearing mice and its transcription is controlled by HIF-1α [144]. Since HIF-1α and AR expression are entwined, it is possible that the expression and function of Rab27a could be altered by targeting ARs.

Although ADO and AR are tightly connected to the regulation of membrane lipids [146], little is known about their involvement in EV production. A study investigating the effects of ARs on exosome production revealed that exosome production in rat preglomerular vascular smooth muscle cells (PGVSMCs) lacking A_1_R and A_2A_R was increased relative to that of wild-type cells under both normal and energy-depleted conditions [147]. The number of exosomes produced by A_2B_R^-/-^ rat PGVSMCs increased under metabolic stress but not under basal conditions, suggesting that A_2B_R is important for exosome release under pathological changes. Conversely, treatment with an A_2B_R-specific antagonist limited exosome production in HNSCC cells after exposure to stress stimuli. In contrast to the effects of A_2B_R antagonism in the same cell line, these findings showed that A_2A_R antagonism upregulated exosome formation in the presence of metabolic inhibitors. In fact, under basal conditions, an A_2A_R-selective agonist even reduced exosome numbers, probably because the receptors were not yet saturated. This effect of A_2A_R stimulation was demonstrated in multiple tumour cell lines, indicating that it represents a common paradigm in exosome regulation. Importantly, exosome release was also stimulated in HNSCC cells by the chemotherapeutic agent cisplatin, whose effects on exosome production resembled those of induced metabolic stress conditions [147]. Interestingly, the knowledge of A_3_R involvement in EV production is limited.

The observations listed above indicate that blocking A_2A_R in anticancer immunotherapy could upregulate EV production and thus act against the goal of the treatment. In addition, the role of A_2B_R in EV-exerted functions seems to be highly context-dependent, indicating that the exact benefits of targeting EVs in tumours still need to be evaluated. 

### 4.6. Prospective Targets of Adenosinergic Therapy

We have investigated signal transduction pathways used in AR-targeting anticancer immunotherapy and other areas that were previously not often considered in the context of the inner signalling of cancer cells [13,148]. Studies in this field have revealed a few molecules with particularly strong connections to the adenosinergic pathway that seem worthy of attention for future efforts to develop effective anticancer ADO therapies (Figure 3).

Bruton’s tyrosine kinase (BTK) is an essential kinase for B cell maturation and signalling whose phosphorylation triggers intracellular signalling via AKT and NF-κB, with Ca^2+^ acting as a second messenger [149]. Differing effects of ADO on regulatory (Breg) and effector (Beff) B cells have been reported in HNSCC [150]. Whereas a subset of Breg cells displayed CD73 ectonucleotidases on their surface to generate eADO, Beff that did not express CD73 utilized eADO in an A_2A_R-dependent manner to downregulate BTK phosphorylation and thus deactivate Beff cells. Interestingly, if A_2A_R was blocked, more B cells infiltrated the tumour, ultimately leading to alleviation of the tumour burden in a murine model. In addition to their role in B cell malignancies, other BTK isoforms were recently identified to be expressed in epithelial cancers [151]. Moreover, BTK was established as a key regulatory kinase of chemokine-controlled migration and B cell functions involving chemokine SDF-1 and chemokine receptors CXCR4 and CXCR5 [149,152]. Since ARs could modulate BTK function in B cells, the finding that BTK is also expressed in epithelial cancers together with other evidence linking ARs to chemokine receptor-regulated migration of cancer cells [110], it remains to be seen whether ARs could also shape BTK-chemokine signalling in solid tumours.

Elevated levels of the metabolite N-acetylaspartate and the enzyme responsible for its production, N-acetylaspartate synthetase (NAT8L), have been observed in non-small cell lung cancer (NSCLC) and advanced ovarian tumours [153,154]. More recently, analysis of DNA methylation in the NAT8L gene was shown to have prognostic value for patients [155]. In addition, concentrations of N-acetylaspartate produced by ovarian cancer cells were found to increase in parallel with cancer progression and correlated with, the polarization of macrophages towards the M2-like phenotype [156]. Non-specific stimulation of ARs by the agonist NECA upregulated the expression of NAT8L RNA in a colitis-associated tumorigenesis mouse model [157]. Because CD73 inhibitor conversely downregulated NAT8L expression, the observations were ascribed to AR modulation. As pointed out in the previous sections, the roles of ADO-ARs’ in cancer metabolism are complex and could also involve crosstalk with NAT8L.

Other interesting partners of AR signalling are ATP-binding cassette (ABC) transporters, which are pivotal for drug efflux as well as for the cytoskeletal rearrangements and high motility of some cancer cells. Evidence was recently presented for an interaction between A_3A_R and P-gp [158], and previous studies indicated that ADO analogues interact with the ABC transporter axis [120,159]. ATP-binding cassette subfamily C member 6 (ABCC6) represents another bridge between xenobiotic transporters and purinergic signalling in HepG2 cells; it acts to increase extracellular reserves of ATP [160]. Moreover, CD73 or ABCC6 inhibition disrupted the filopodia architecture in HepG2, whereas ADO addition preserved it, indicating the favourable interaction between ABCC6 and ADO signalling. Interestingly, CD73 expression was diminished when the ABCC6 gene was knocked down in HepG2 cells [161], and strong CD73 expression led to enhanced expression of EGFR; together, these changes favour HCC growth and motility [162]. Since the non-enzymatic pro-tumour function of CD73 is known [163], careful evaluations will be needed to determine whether the impact of the adenosinergic pathway on ABCC6 occurs exclusively via ARs.

Previous studies showed that A_2A_R was upregulated in murine T cells via T cell receptor and nuclear factor of activated T cells (NFAT) [164]. Moreover, stimulation of A_2A_R by CGS21680 led to deregulation of NFAT in Jurkat T cells via the PKA axis [165]. The function of NFAT is also modulated by several proliferation stimuli that are controlled by ARs, including upstream kinases and intracellular Ca^2+^ flux [166]. NFAT is an important regulator of cellular proliferation, angiogenesis, motility, and inflammation. In CD8+ Teff cells, NFAT also regulates the expression of a receptor for integrin αE (CD103), which forms a heterodimer with integrin β7 that binds to E-cadherin [167]. Alongside CD8+ Teff cells [168], an A_2A_R-specific antagonist rescued CD103+ antigen-presenting DCs from ADO-induced immunosuppression [42]. Collectively, these results demonstrate a multilevel connection between NFAT and ADO in immune cells that could analogously be hijacked by cancer cells.

AR modulators seem to be tightly associated with the glucose-regulated protein (GRP) family. Earlier studies demonstrated activation of the ER stress response by ADO and its analogues [169]. ADO treatment upregulated GRP78 expression and caused ER stress-triggered apoptosis in oesophageal cancer cells in vitro [170]. Later, the same research group reported that knockdown of GRP78 facilitated anti-tumour effects of ADO in HepG2 cells [171]. In addition, the non-selective AR agonist NECA directly bound and inhibited GRP94, another GRP member localized in the ER [172]. Interestingly, a growing body of evidence suggests that ER stress signals regulate various downstream pathways (PI3K/AKT, NF-κB, MAPK/ERK, TGF-β, and Wnt/β-catenin) and that these effects are mediated by GPCRs; for a summary, see [173]. Since GRPs are crucial regulators of ER functions, the clarification of their relationship to ARs and their ligands could be important for AR drug discovery.

There is currently little data on the role of the ADO pathway in regulating adhesion molecules such as the intercellular adhesion molecule 1 (ICAM-1) either involving ARs or intracellular ADO activation of the transmethylation pathway. However, activation of A_2B_R in endothelial cells [174] and A_2A_R in human monocytes inhibited expression of ICAM-1 by upregulation of cAMP; conversely, inhibition of A_2A_R or activation of A_1_R or A_3_R led to increased levels of ICAM-1 [175]. ADK is activated during inflammation, causing depletion of intracellular ADO, stimulation of histone methylation, and upregulation of adhesion molecules in endothelial cells [176]. In addition to ADK blockage, cellular intake of excess eADO blocks the transmethylation pathway (Figure 1) and thus reduces the expression of ICAM-1 and other adhesion molecules. While this is beneficial for vascular and other inflammatory conditions, it may be unfavourable in cancer. In this context, it is notable that circulating tumour cell (CTC) clusters exhibit altered adhesion molecule profiles and ICAM-1 is one of the adhesion molecules that mediates aggregation of CTCs. The significance of multicellular CTC clusters for cancer patients was recently reviewed [177]. It should also be noted that the heterotypic CTC clusters include immune cells and CAFs and that their escape from the primary tumour site is enabled by leaky vasculature [177]. Finally, DNA methylation is an essential factor for stemness (re-)programming of CTCs [178]. Taken together, we cannot exclude the impact of adenosinergic therapy on CTCs.

## 5. Persisting and Potential Limitations of Adenosinergic Therapy

There are several potential limitations of adenosinergic therapy arising from the fact that ARs have the GPCR structure and from their relationships with other molecules. In this section, we summarized the factors that contribute to the complexity of adenosinergic signalling.

### 5.1. Structure-Related Limitations

ARs belong to class A GPCRs and are thus primarily regulated by ligand binding to an extracellular binding site [179,180]. The conformational rearrangements caused by ligand binding lead to (I) monomer receptor signalling in the cell membrane, (II) receptor oligomerization, (III) receptor engagement in supramolecular assemblies in the cell membrane (receptor–receptor interaction, RRI), (IV) transactivation without physical interaction, and (V) signalling beyond the boundary of the cell membrane (e.g., via exosomes and EVs). The complexity of AR signalling gives rise to several limitations that must be overcome to develop effective therapies. The recently formulated concept of GPCRs clustering with other receptors to form supramolecular assemblies invites novel strategies against cancer [180]. Importantly, this concept goes beyond RRI and implies the involvement of other accessory proteins such as receptor tyrosine kinases, scaffold proteins, or ion channels. The RRI can indeed change the context of GPCR signal transduction, as demonstrated by the fact that GPCR heterodimerization can lead to altered G protein preferences in the endocrine system [181,182], neurophysiology [183,184,185,186,187], and tumours [188].

A recent study on ‘megaplexes’ [189] explained how internalized GPCRs retain their signalling while bound to endosomes and revolutionized the perception of GPCR-agonist signalling. This report also highlighted the complexity of GPCR signalling. Another factor limiting the functional activity of ARs is their mutations, as demonstrated forA_2B_R [190]. A mechanistic study of A_2B_R mutants in yeast provided both constitutively active and inactive mutants of A_2B_R [191]. Furthermore, A_2B_R could be constitutively active in prostate cancer independently of ligand binding [192]. The presence of cancer-related somatic mutations could explain seemingly contradictory findings of the role of A_2B_R and possibly other ARs in cancer. Moreover, an essential hindrance in AR knowledge is the lack of A_2B_R and A_3_R crystal structures to deepen our understanding of A_2B_R and A_3_R conformational dynamics.

Targeting ARs often leads to interactions with multiple transduction pathways, which is desirable under certain conditions. However, AR modulation can also cause signalling to be biased towards certain pathways that are more precise and suffer less from adverse effects [179,193,194]. Interestingly, though, stimulation with AR modulators, eADO, or ADO-derived compounds was reported to often have the same anticancer effects although mediated via different pathways, including inducing cell cycle arrest and targeting VEGF [148]. 

The ARs are in general pharmacologically modulated by small molecules and the engagement of multiple targets (so-called polypharmacology) is an existing disadvantage of small molecules. Accordingly, a number of compounds derived from ADO structure initially identified as AR-interacting partners were later reported to have polypharmacological effects. For example, using a target deconvolution study, Yu et al. found multiple binding partners of IB-MECA, which was previously described as a selective A_3_R agonist [195]. In addition, several compounds that were first identified as having intracellular targets have since been confirmed to be AR-binding partners [148,196]. Finally, the selective A_2B_R agonist BAY 60-6583 was recently shown to engage another target molecule to upregulate CAR-T cell activity independently of A_2B_R [197]. The possibility that ADO analogues may have multiple targets therefore cannot be excluded without careful evaluation. In addition to the polypharmacology of AR modulators, the ECL2 (extracellular loop 2) influences the stability and kinetics of ligand binding of ARs [198,199]. A study using a chimeric human A_2A_R containing the extracellular loop 2 (ECL2) from A_2B_R, A_2A_ (ECL2-A_2B_)AR, identified this region to be crucial for its affinity for ADO and ADO potency on A_2A_R [200]. These effects, however, did not apply to the synthetic compound NECA and its derivative CGS-21680, highlighting how important minute structural differences can be.

### 5.2. Context-Related Limitations

Targeting the adenosinergic pathway in cancer could work bidirectionally. The currently prevalent view links increased CD73 expression to poor clinical outcomes; accordingly, several anticancer therapies targeting CD73 are underway [7,13]. However, Bowser et al. showed loss of CD73 to be essential for endometrial tumour progression [201], while Kurnit et al. reported the critical role of CD73 in the tumour-suppressive activity of TGF-β1 in endometrial carcinoma [202]. Both studies also associated actin polymerization with the CD73-A_1_R axis, and Kurnit et al. provided evidence for a CD73-TGF-β negative feedback loop in which the A_1_R-selective agonist CPA reduces TGF-β1-mediated invasion of HEC-50 cells [202]. Because micromolar concentrations of CPA were used, it remains to be seen whether this effect is exclusively A_1_R-dependent. Taking the results presented in the preceding sections into account, it must be noted that while blocking CD73 may prove beneficial in some tumours, it could contribute to the acceleration of tumour progression in others. 

The immunosuppressive effects of eADO were ameliorated by a novel dual A_2A_R/A_2B_R antagonist SEL330-639 [203]. Additionally, it was shown that A_2B_R has a higher affinity for ADO in HEK293 cells overexpressing A_2B_R than was previously reported and that the longer residence time of the antagonist impacted the outcome to a greater degree than differences in affinity for individual receptors, especially in cases where the antagonists compete with high levels of eADO as in the TME. Interestingly, a mechanistic study identified A_2A_R as a determinant of A_2B_R expression [204]. Furthermore, SCH58361, an A_2A_R antagonist, re-activates CD8+ Teff cells and stimulates the population of inflammatory M1-like macrophages in chronic lymphocytic leukaemia [205]. The effectiveness of anti-PD-L1 therapy is reduced by strong eADO expression but can be restored by blocking eADO and A_2A_R [13,40]. Despite these benefits of blocking A_2A_R in cancer immunotherapy, it suffers from an unexpected limitation. Even if the A_2A_R inhibitor re-sensitizes immune Teff cells in TME, the immune exhaustion could be irreversible, limiting the application of AR modulators in immunotherapy-resistant tumours such as those with dominant A_2A_R/CD73/CD39 axis [206]. In contrast, stimulation of A_2A_R/A_2B_Rs reduces TGF-β-modulated contractility and migration of mammary fibroblasts, both of which are essential for ECM remodelling and tumour metastasis facilitated by CAFs [207]. ADO for activation of A_2A_R/A_2B_Rs in fibroblasts was provided by nearby CD73^+^ myeloid cells [106]. In addition, an analysis of breast cancer survival data by Vasiukov et al. revealed a positive correlation between A_2A_R gene expression and better outcomes in patients with basal type and TNBC. In contrast, A_2B_R levels correlated negatively with overall survival [106]. ADO was previously reported to have both stimulatory and inhibitory effects on melanoma cells [208], and several studies have reported contradictory effects of ARs on tumour cells [64]. These reports imply strong dependency on the context. 

Since excessive extracellular levels of ADO are undesired in the TME, the ADO-degrading enzymes ADA and PNP which could shift the balance toward ADO consumption and utilization (Figure 1), could be future targets for anticancer therapy [7]. However, inosine, a product of ADO deamination, functions as an AR agonist with a longer half-life than ADO and exerts its anti-inflammatory response in mice by activating both A_3_R and A_2A_R [209,210]. Thus, whereas eliminating eADO by metabolizing it could lead to cancer cell death, its consequences might still compromise immune cell response in the TME. 

Moreover, it has been recently postulated that the host immune system is an essential factor for the success of chemotherapy [211]. Current consensus associates the ‘oncobiome’ with patient survival and with the adverse events of chemo- and immunotherapy because of the beneficial effects of the commensal bacteria on immune cells in the gut. The intestine microbiome was only recently shown to stimulate T helper cell type 1 (Th1) cells via the production of inosine (Figure 2C). Although inosine is predominantly associated with immunosuppressive actions similar to eADO, in this case, inosine stimulate naïve T cells toward Th1 via A_2A_R-cAMP-PKA-pCREB circuitry specific for intestinal T-cells and thus enhances the efficacy of ICI therapy [212,213]. This unique inosine-A_2A_R signalling is context-dependent and requires co-stimulation. So far, the benefits of A_2A_R antagonists in immune-oncology seem to outweigh the negatives, but the favourable effect of inosine in this study was entirely abrogated by the A_2A_R-specific antagonist ZM241385 [212]. Seeing from a greater perspective, more studies are required to fully understand the holistic impact of adenosinergic therapy.

## 6. Perspectives

Despite the clarification of many aspects of the adenosinergic system over the past two decades, several questions remain. We summarise some of them in Table 2. Novel experimental models and promising approaches may assist in answering these questions in the future.

Jacobson and Reitman have sensibly called for further knock-out in vivo studies in order to support novel findings and re-evaluate some that were reported previously [214]. On the other hand, the question of data translatability from animal studies for human therapy persists. Therefore, in addition to employing rodent models, other novel approaches that are more considerate to animals are desired. For instance, organoid tumour models lacking one or all four ARs would be equally challenging and beneficial for further investigations of ARs’ role in cancer, and for the mapping of reciprocal complementarity of key ADO-generating enzymes. In addition, the compensatory mechanisms of the adenosinergic orchestra could be better interrogated using gene-editing CRISPR/Cas9 technology [215,216].

A study targeting A_2A_R showed that the effects of shRNA silencing may differ markedly from those of treatment with a specific antagonist targeting the silenced protein. shRNAs against A_2A_R expressed by a CAR construct promoted proliferation, cytokine production, and cytotoxicity of anti-mesothelin CAR-T cells toward cancer cells. Conversely, A_2A_R inhibition by a specific antagonist did not induce the desired cytotoxic activity of CAR-T cells [44], suggesting either a diversity in this protein’s mechanism of action or insufficient biodistribution of the pharmacological agent. In this context, it is important to note that the effects of eADO are spatially and temporally restricted and that the ADO-enriched tumour niche is not readily accessible to many therapeutics [7]. Despite these challenges, it would be highly desirable to localize adenosinergic therapies to solid tumours in order to avoid wide-ranging unspecific engagement of eADO and ARs. Novel drug delivery systems and controlled release of therapeutics are therefore drawing increasing attention, in part because of immune-related adverse events that might occur due to re-acquired or enhanced tumour response upon immuno-stimulatory therapy [217,218]. Nanoparticles have already been successfully used to experimentally knock down A_2A_R in CD8+ Teff cells and to thereby restore their functions [219,220]. Other nanosystems are in development to improve compound delivery to AR-overexpressing locations [221] or to study membrane proteins such as ARs under more native conditions that better resemble the cell membrane’s structure [222]. Moreover, it is becoming increasingly clear that EVs play key roles in cancer adenosinergic signalling. Currently, the therapeutic potential of engineered myeloid stem/stromal cell-derived EVs with defined cargo (proteins, siRNAs, miRNAs, nanobodies or encapsulated chemotherapeutic agents) as a convenient drug delivery system is expanding [223,224] and could be further harnessed to target the adenosinergic system specifically.

Complementing targeted delivery systems, novel cheminformatics approaches could clarify the binding of ligands to ARs, as demonstrated for A_2A_R [225], and could also reveal key structural differences between agonistic or antagonistic behaviour of compounds, as shown for A_3_R [226]. Rapid advances in structure-based drug discovery may enable the generation of higher resolution AR structures [227]. The use of structure-based drug design can help reveal mechanistic details of ligand-receptor binding to avoid unexpected pitfalls and possible adverse effects earlier in the process, facilitating advancements in GPCR drug discovery [228]. Finally, quantitative mathematical modelling could be used to assess the effectiveness of combined treatment strategies based on preclinical data [229].

## 7. Conclusions

The ADO-rich TME promotes immunosuppression and metabolic reprogramming of immune cells at the tumour site. We have summarised recent research illustrating the pro- and anti-tumour roles of the adenosinergic orchestra and the possible consequences of its targeting for different cell types, with emphasis on cancer cells, highlighted factors that may complicate its clinical targeting, and evaluated prospective targets within the adenosinergic pathway. There are clear spatial and temporal patterns of ADO influence, as well as persistent limitations to ADO-targeting therapeutic strategies that include the polypharmacology of ADO analogues and the lack of crystal structures for A_2B_R and A_3_R. We concluded by offering some suggestions for future directions in ADO-AR research.

## Figures and Tables

**Figure 1 ijms-22-12569-f001:**
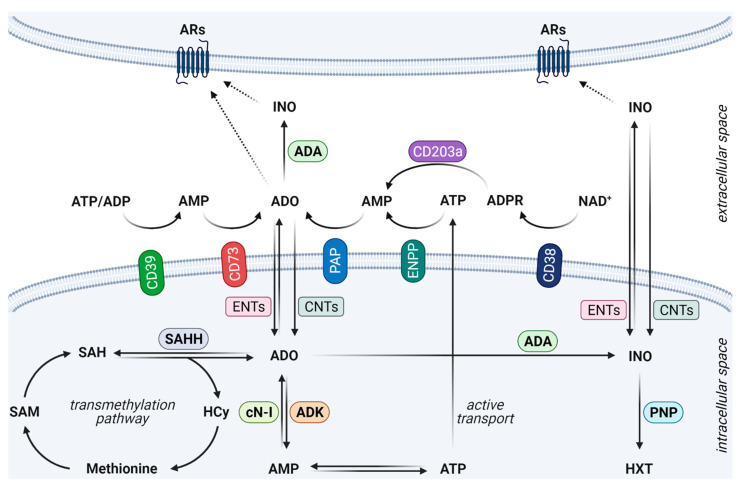
An overview of ADO production, metabolism, transport and signalling. ATP is actively transported from cells by the non-lytic mechanisms (connexin and pannexin hemichannels and other transporters) or uncontrollably released after stress stimuli. Extracellular ATP is hydrolysed by CD39 and CD73 ectonucleotidases to ADO. ADO could be also produced by ENPP and PAP enzymatic activity or alternatively by CD38 from NAD^+^. ADO is further metabolised to INO by ADA, and INO is converted to HXT by PNP. ADO is also important for the transmethylation pathway, and its intracellular availability is regulated by SAHH, ADK and cN-I. ADO and INO could be transported by ENTs (both directions) and CNTs (one-way transport). In the extracellular space, both ADO and INO interact with ARs in an autocrine and paracrine manner. ADA, adenosine deaminase; ADK, adenosine kinase; ADO, adenosine; ADP, adenosine diphosphate; ADPR, adenosine diphosphate ribose; AMP, adenosine monophosphate; AR, adenosine receptor; ATP, adenosine triphosphate; cN-I, cytoplasmic 5′-nucleotidase-I; CNT, concentrative nucleoside transporter; ENPP, ectonucleotide pyrophosphatase/phosphodiesterase; ENT, equilibrative nucleoside transporter; HCy, homocysteine; HXT, hypoxanthine; INO, inosine; NAD+, nicotinamide adenine dinucleotide; PAP, prostatic acid phosphatase; PNP, purine nucleoside phosphorylase; SAH, S-adenosylhomocysteine; SAHH, S-adenosylhomocysteine hydrolase; SAM, S-adenosylmethionine.

**Figure 2 ijms-22-12569-f002:**
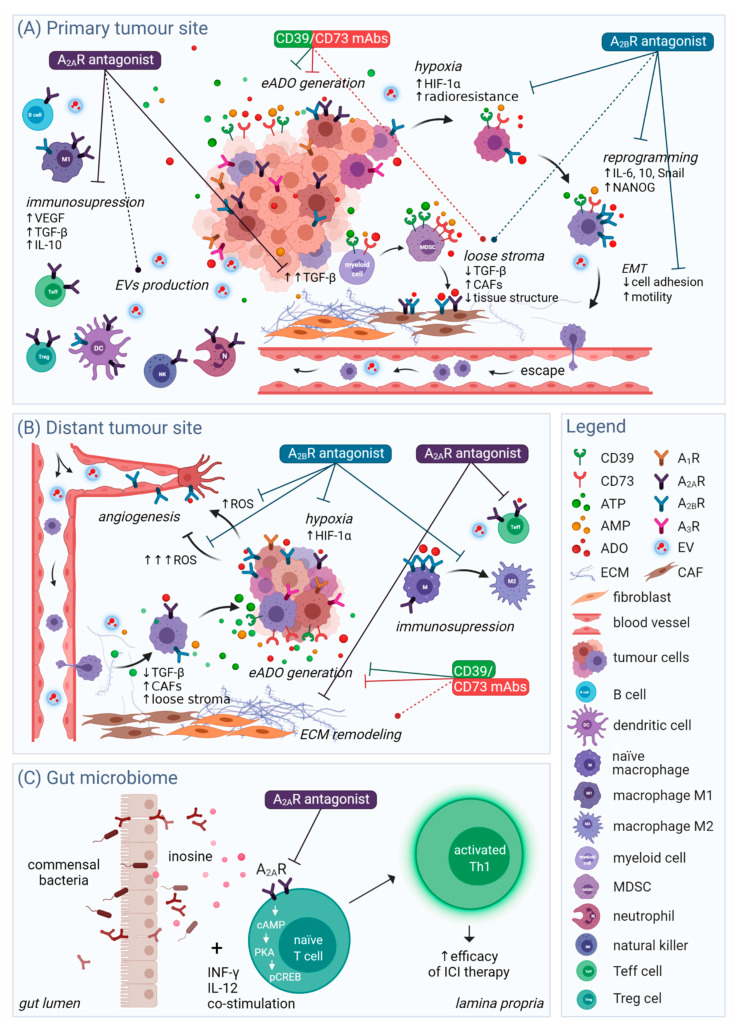
Selected aspects of the multilevel impact of novel adenosinergic therapies. (**A**) At the primary tumour site and (**B**) distant tumour site, the blockage of CD39/73 by mAbs attenuates the generation of eADO from its precursor ATP. Simultaneously, CD73 exerts an ambiguous role in tissue structure maintenance and stroma remodelling (dashed line). Targeting of the A_2A_R transduction pathway with small-molecule antagonists results in inhibition of ADO-mediated immunosuppression and production of TGF-β, which is necessary for maturation of myeloid cells into MDSC. Blockage of A_2A_R positively affects EVs production, as suggested by preliminary reports. Antagonism of A_2B_R attenuates the effects of hypoxia-driven tumour progression and radioresistance while downregulating cellular reprogramming and the EMT process. Additionally, A_2B_R modulates TGF-β production in a manner that depends on cell type and related factors and thus has an ambiguous effect on tissue structure. Blocking A_2B_R also inhibits the polarization of macrophages to the immune-tolerant M2-like phenotype and reduces ROS generation. Controlled production of ROS stimulates angiogenesis, whereas ROS overproduction causes detrimental oxidative stress in endothelial cells leading to cell death. A_2B_R could thus have both proangiogenic and anti-angiogenic effects. (**C**) Commensal bacteria in the gut release inosine into the lamina propria, which stimulates the differentiation of naïve T cells into Th1 in an A_2A_R-dependent manner specific to intestinal T cells. Activated Th1 cells then facilitate ICI therapy. Therefore, while exerting anticancer effects at the tumour site, the inhibition of A_2A_R could potentially limit the effectiveness of immuno-therapy. A_1_R, adenosine A_1_ receptor; A_2A_R, adenosine A_2A_ receptor; A_2B_R, adenosine A_2B_ receptor; A_3_R, adenosine A_3_ receptor; AMP, adenosine 5′-monophosphate; ATP, adenosine 5′-triphosphate; CAF, cancer-associated fibroblast; eADO, extracellular adenosine; ECM, extracellular matrix; EMT, epithelial-mesenchymal transition; EV, extracellular vesicle; HIF-1α, hypoxia-inducible factor 1α; ICI, immune checkpoint inhibitor; IL-10, interleukin-10; IL-6, interleukin-6; mAb, monoclonal antibody; MDSC, myeloid-derived suppressor cell; ROS, reactive oxygen species; Teff, T effector cell; TGF-β, transforming growth factor β; Th1, T helper cell type 1; Treg, T regulatory cell; VEGF, vascular endothelial growth factor.

**Figure 3 ijms-22-12569-f003:**
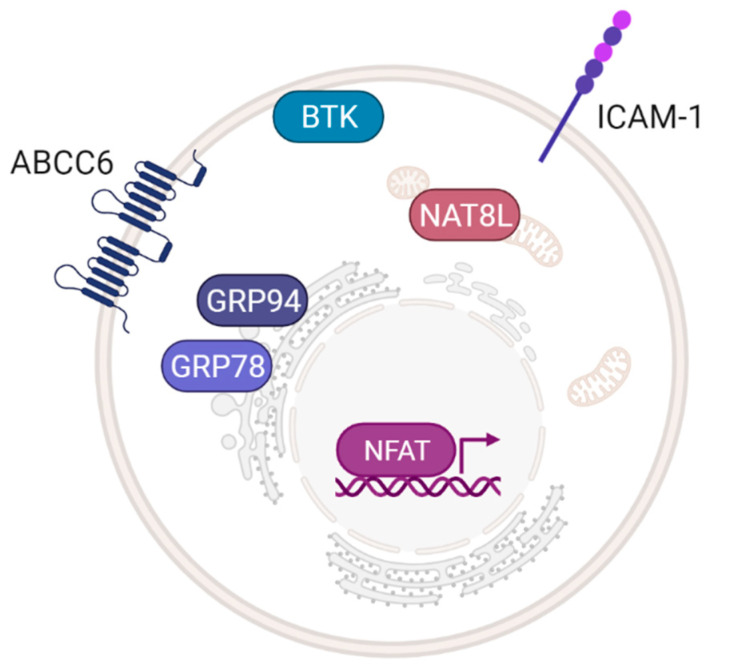
Prospective targets of the adenosinergic pathway in tumours. ABCC6, ATP-binding cassette subfamily C member 6; BTK, Bruton’s tyrosine kinase; GRP78, glucose-regulated protein of 78 kDa; GRP94, glucose-regulated protein of 94 kDa; ICAM-1, intercellular adhesion molecule 1; NAT8L, N-acetylaspartate synthetase; NFAT, nuclear factor of activated T cells.

**Table 1 ijms-22-12569-t001:** Clinical trials targeting the adenosinergic pathway in malignancies initiated between 2020 and 2021 (source: ClinicalTrials.gov, accessed 30 September 2021).

NCT Number	Target	Type of Agent	Agent	Combination Therapy	Condition	Phases
NCT04280328	A_2A_R	Antagonist	Ciforadenant (CPI-444)	Daratumumab (CD38)	Relapsed or refractory MM	I
NCT04381832	A_2A_R/A_2B_R	Dual antagonist	AB928 (etrumadenant)	Zimberelimab (PD-1) ± enzalutamide (androgen receptor), docetaxel or AB680 (CD73) ± zimberelimab (PD-1)	Metastatic castrate resistant prostate cancer	I/II
NCT04660812	A_2A_R/A_2B_R	Dual antagonist	AB928 (etrumadenant)	Zimberelimab (PD-1) ± mFOLFOX6, bevacizumab (VEGF), regorafenib (kinases inhibitor)	Metastatic CRC	I/II
NCT04017130	CD38	ETB targeting CD38	TAK-169	-	Relapsed or refractory MM	I
NCT04083898	CD38	IgG1 anti-CD38 mAb	Isatuximab	Bendamustine, prednisone	Relapsed or refractory MM	I/II
NCT04352205	CD38	IgG1 anti-CD38 mAb	Daratumumab	Bortezomib, dexamethasone ± thalidomide or lenalidomide	MM with renal failure	II
NCT04430530	CD38	CAR-T	4SCAR-T specific to CD22/CD123 /CD38/CD10/CD20	-	CD19 negative B-cell malignancies	I/II
NCT04270409	CD38	IgG1 anti-CD38 mAb	Isatuximab	Lenalidomide, dexamethasone	Smoldering MM	III
NCT03841565	CD38	IgG1 anti-CD38 mAb	Daratumumab	Pomalidomide, dexamethasone	Relapsed MM	II
NCT04251065	CD38	IgG1 anti-CD38 mAb	Daratumumab	Gemcitabine, cisplatin, dexamethasone	Relapsed or refractory T-cell lymphoma	II
NCT04230304	CD38	IgG1 anti-CD38 mAb	Daratumumab	Ibrutinib (BTK inhibitor)	Relapsed or refractory chronic lymphocytic leukaemia	II
NCT04566328	CD38	IgG1 anti-CD38 mAb with hyaluronidase	Daratumumab and hyaluronidase-fihj	Lenalidomide, dexamethasone ± bortezomib	MM	III
NCT04316442	CD38, tubulin polymerization	Antibody-drug conjugate of anti-CD38 mAb and duostatin 5.2	STI-6129	-	Relapsed or refractory systemic AL amyloidosis	I
NCT04407442	CD38	IgG1 anti-CD38 mAb	Daratumumab	Azacitidine, dexamethasone	Relapsed or refractory MM	II
NCT04150692	CD38	IgG1 anti-CD38 mAb with hyaluronidase	Daratumumab and hyaluronidase-fihj	-	Relapsed or refractory MM	II
NCT04824794	CD38	IgG1 anti-CD38 mAb	GEN3014 (HexaBody-CD38)	-	Relapsed or refractory MM	I/II
NCT04758767	CD38	IgG1 anti-CD38 mAb	CID-103	-	Relapsed or refractory MM	I
NCT04139304	CD38	IgG1 anti-CD38 mAb	Daratumumab	DA-EPOCH	Plasmablastic lymphoma	I
NCT04802031	CD38	IgG1 anti-CD38 mAb	Isatuximab	-	Relapsed or refractory MM	II
NCT04892264	CD38	IgG1 anti-CD38 mAb	Daratumumab	Belantamab (BCMA), mafodotin (microtubule inhibitor), lenalidomide	Untreated, relapsed or refractory MM	I/II
NCT04763616	CD38	IgG1 anti-CD38 mAb	Isatuximab	Cemiplimab (PD-1)	Relapsed or refractory NK/T-cell lymphoid malignancy	II
NCT05011097	CD38, CD3	Anti-CD38 and anti-CD3 bispecific antibody	Y150	-	Relapsed or refractory MM	I
NCT04751877	CD38	IgG1 anti-CD38 mAb	Isatuximab	Lenalidomide and dexamethasone ± bortezomib	MM	III
NCT04336098	CD39	Anti-CD39 mAb	SRF617	±gemcitabine + paclitaxel or pembrolizumab (PD-1)	Advanced solid tumours	I
NCT04306900	CD39	Anti-CD39 mAb	TTX-030	mFOLFOX6, docetaxel, nab-paclitaxel, gemcitabine and/or budigalimab (PD-1) or pembrolizumab (PD-1)	Advanced solid tumours	I
NCT04672434	CD73	anti-CD73 mAb	Sym024	±Sym021 (PD-1)	Advanced solid tumours	I
NCT04668300	CD73	IgG1 anti-CD73 mAb	Oleclumab	Durvalumab (PD-L1)	Recurrent, refractory, or metastatic sarcoma	II
NCT04262375 †	CD73	IgG1 anti-CD73 mAb	Oleclumab	Durvalumab (PD-L1)	Advanced NSCLC or RCC	II
NCT04262388 †	CD73	IgG1 anti-CD73 mAb	Oleclumab	Durvalumab (PD-L1)	PDAC, NSCLC and HNSCC	II
NCT04776018 *	SUMOylation	Small molecule inhibitor	TAK-981 (subasumstat)	Mezagitamab (CD38) ± daratumumab and hyaluronidase-fihj (CD38)	Relapsed or refractory MM	I/II
NCT05060432 *	TIGIT	IgG1 anti-TIGIT mAb	EOS-448	Pembrolizumab (PD-1) or inupadenant (A_2A_R)	Advanced solid tumours	I/II
NCT04205240 *	-	allo HSCT	-	Cyclophosphamide, fludarabine, melphalan; mycophenolate mofetil, tacrolimus (immunotherapy); daratumumab (CD38)	Relapsed MM	II

Abbreviations: 4SCAR-T, 4th generation chimeric antigen receptor gene-modified T cells; allo HSCT, allogenic hematopoietic stem cell transplantation; BCMA, B cell maturation antigen; BTK, Bruton’s tyrosine kinase; CAR-T, chimeric antigen receptor T cells; CRC, colorectal cancer; DA-EPOCH, dose-adjusted etoposide, prednisone, vincristine sulfate, cyclophosphamide, and doxorubicin hydrochloride; ETB, engineered toxin body; HNSCC, head and neck squamous cell carcinomas; MM, multiple myeloma; mAb, monoclonal antibody; NK, natural killer; NSCLC, non-small cell lung cancer; PD-1, programmed cell death protein 1; PD-L1, programmed cell death protein ligand 1; PDAC, pancreatic ductal adenocarcinoma; RCC, renal cell carcinoma; TIGIT, T cell immunoreceptor with Ig and ITIM domains. * adenosinergic therapy as a secondary target or co-therapy, † withdrawn.

**Table 2 ijms-22-12569-t002:** Important questions about the adenosinergic pathway.

1	Based on current knowledge, intracellular ADO triggers epigenetic reprogramming independently of ARs. Low levels of intracellular ADO can boost DNA methylation, whereas its accumulation blocks epigenetic changes [9]. Could there be a direct feed-forward loop between intracellular ADO and AR expression?
2	When the therapy targets ADO-rich tumours and blocks A_2A_R on immune cells by A_2A_R antagonist for instance, what happens to the excessive eADO in the niche? Could continuously generated ADO backfire as a result? Will the ADO metabolites engage other pro-tumoral molecular processes? What pathways will be heightened?
3	How to better understand the inconsistencies of adenosinergic pathways in different tumour models?

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
