# Peer review of "Current Adenosinergic Therapies: What Do Cancer Cells Stand to Gain and Lose?"

_ijms, 2021, doi:10.3390/ijms222212569_

Round 1

Reviewer 1 Report

The present review from Kotulova et al entitled “current adenosinergic therapies: what do cancer cells stand to gain and lose?” is well written and organized. The paper contains all the latter information in the field, and adequately describes the recent research reporting the pro and anti-tumoral roles of the adenosinergic system, the effects on cancer cells of the targeting adenosine receptors, evaluating pro and cons of targets within the adenosinergic pathway.

I think that the paper is publication-worthy in the International Journal of Molecular Sciences in the present form.

Author Response

We are delighted to see the positive opinion of this Reviewer that the paper is publication-worthy in the IJMS in the present form.

Reviewer 2 Report

This is an interesting, relevant and ambitious work on the multiple mechanisms in which the adenosinergic system may influence the tumor environment and on the impact of alterations in this system may have on tumor evolution.
The work has the merit of presenting in an exhaustive way, the possible mechanisms. However, it is very dense and its reading will not be easy for non-specialists.
The organization could be improved. For instance, Section 4 “Targeting ARs on cancer cells” is divided into subsections in which the authors intend to “...summarise recent findings concerning direct and indirect effects of targeting ARs in the tumor niche on various aspects of tumorigenesis”. The following subsections are “Cancer cell proliferation”, “Hypoxia and immunomodulation”, “Migration and angiogenesis”, “Tumour cell stemness and reprograming”, “Extracellular vesicles” and “Prospective targets of adenosinergic therapy”. The extracellular vesicles are discussed in subsection 4.3 when only point 4.5 explains their relevance to tumorigenesis.
The diversity of AR-mediated effects is hard to follow. After section 4, the reader will expect to find in section 5 (Persisting and potential limitations of adenosinergic therapy) a safe haven to consolidate the fragmented information from the previous section. However, it is a false hope. The authors begin by presenting an “Important questions about the adenosinergic pathway (Table 2)” which would meet the reader's expectations, BUT the following subsections are not aligned with the issues and the work would improve if the authors made an extra effort to achieve more consistency in this alignment.

Other points:
Lines 100 – 103: - it is difficult to understand the connection of this information about P. aeruginosa with the rest of the text.
Line 415: the effects of pentoxifylline as an A2B AR should also be taken in consideration
Lines 446 – 450: please rephrase. Could not understand what the authors want to mean!
Lines 480 – 482: hard to understand. Please rephrase!

It is used a huge set of abbreviations that make reading difficult. The presentation of a list of abbreviations would help.

Author Response

Reviewer 2:

This is an interesting, relevant and ambitious work on the multiple mechanisms in which the adenosinergic system may influence the tumor environment and on the impact of alterations in this system may have on tumor evolution.

The work has the merit of presenting in an exhaustive way, the possible mechanisms. However, it is very dense and its reading will not be easy for non-specialists.
The organization could be improved. For instance, Section 4 “Targeting ARs on cancer cells” is divided into subsections in which the authors intend to “...summarise recent findings concerning direct and indirect effects of targeting ARs in the tumor niche on various aspects of tumorigenesis”. The following subsections are “Cancer cell proliferation”, “Hypoxia and immunomodulation”, “Migration and angiogenesis”, “Tumour cell stemness and reprograming”, “Extracellular vesicles” and “Prospective targets of adenosinergic therapy”. The extracellular vesicles are discussed in subsection 4.3 when only point 4.5 explains their relevance to tumorigenesis.

The diversity of AR-mediated effects is hard to follow. After section 4, the reader will expect to find in section 5 (Persisting and potential limitations of adenosinergic therapy) a safe haven to consolidate the fragmented information from the previous section. However, it is a false hope. The authors begin by presenting an “Important questions about the adenosinergic pathway (Table 2)” which would meet the reader's expectations, BUT the following subsections are not aligned with the issues and the work would improve if the authors made an extra effort to achieve more consistency in this alignment.

RESPONSE: The major concern of Reviewer 2 is the readability of the text for non-specialists. Therefore, we have made changes to the manuscript to target this issue.

We aimed to summarize in one manuscript the most important aspects where adenosine and adenosine receptors modulate tumour response. The role of adenosine and its receptors in malignancies is indeed a complex topic and we acknowledge that certain background information is necessary to understand all the described phenomena. For these reasons, we direct the non-specialist readers throughout the manuscript to previous works on separate topics (e.g. references 7, 8, 13 – 15, 17, 173, 177).

Also, to target a more general audience, we included an adenosine metabolism scheme (Figure 1).

To improve the readability of chapter 4.6. we have highlighted the prospective targets in the text.

The extracellular vesicles are truly mentioned in 4.3 without any other explanation in that section. We omitted the extracellular vesicles here, as they are not necessary for the given information and elaborated on this study more in section 4.5.

The multi-level structure of the manuscript (from cancer cell proliferation to prospective targets) only mirrors the complexity of the adenosinergic pathway effects. Unfortunately, the information on the adenosine role is fragmented, and the answers are often missing, as it is with our unanswered questions.

However, we moved Table 2 to section 6, as we believe that the “6. Perspectives” section is more related to the questions in Table 2 and, if partially, offers directions for the readers.

Other points:

Lines 100 – 103: - it is difficult to understand the connection of this information about P. aeruginosa with the rest of the text.

The connection of P. aeruginosa to the paragraph as a source of infection here is less relevant to the effects of ENTs on the availability of extracellular adenosine and might possibly confuse the reader (without functional ENT1 there is more extracellular ADO available which results in activation of A2AR and A2BR and subsequent suppression of the excessive inflammatory response to the P. aeruginosa infection). We amended this part and hopefully made it clear now.

Line 415: the effects of pentoxifylline as an A2B AR should also be taken in consideration

We included the information about pentoxifylline with the corresponding reference.

Lines 446 – 450: please rephrase. Could not understand what the authors want to mean!

We amended these lines to clarify the information.

Lines 480 – 482: hard to understand. Please rephrase!

We amended these lines to clarify the information.

It is used a huge set of abbreviations that make reading difficult. The presentation of a list of abbreviations would help.

We acknowledged the extent and complexity of the review and added the list of abbreviations at the end of the manuscript.

Reviewer 3 Report

The manuscript is well written. It can provide timely and detailed knowledge of adenosine and its receptor role in the tumor microenvironment. The reviewer has a few minor comments.

  1. The reviewer suggests including a schema of adenosine metabolism in the section 1. Since this paper is informative, it can be read not only adenosine specialist, but also readers with less knowledge of adenosine.
  2. Table 2 looks not informative for me.
  3. In the line 574, subject is missing in the sentence.

Author Response

Dear Reviewer,

Before listing our detailed point-by-point responses, we wish to emphasize, that we greatly appreciate the constructive comments that helped us to improve further the manuscript, the revised version of which is attached. We have tried to address all issues pointed out by your review and improve the text and understandability of our work. The responses to the particular questions and comments are below.

Reviewer 3:

The manuscript is well written. It can provide timely and detailed knowledge of adenosine and its receptor role in the tumor microenvironment. The reviewer has a few minor comments.

  1. The reviewer suggests including a schema of adenosine metabolism in section 1. Since this paper is informative, it can be read not only adenosine specialist, but also readers with less knowledge of adenosine.

RESPONSE: We acknowledged the suggestion and implemented the adenosine metabolism scheme (Figure 1).

  1. Table 2 looks not informative for me.

RESPONSE: Table 2 highlights the persisting fundamental questions and encourages the readers to consider the impact of the adenosinergic pathway and its targeting from different points of view. For this purpose, we provided a short introduction to Table 2 and moved Table 2 to section 6. Perspectives.

  1. In line 574, subject is missing in the sentence.

RESPONSE: This sentence was amended.